# Antimicrobial Activity of Microorganisms Isolated from Ant Nests of *Lasius niger*

**DOI:** 10.3390/life10060091

**Published:** 2020-06-22

**Authors:** Tatiana A. Efimenko, Alla A. Glukhova, Mariia V. Demiankova, Yuliya V. Boykova, Natalia D. Malkina, Irina G. Sumarukova, Byazilya F. Vasilieva, Eugene A. Rogozhin, Igor A. Ivanov, Vladislav A. Krassilnikov, Olga V. Efremenkova

**Affiliations:** 1Gause Institute of New Antibiotics, 119021 Moscow, Russia; alglukhova@yandex.ru (A.A.G.); mary_bunny@mail.ru (M.V.D.); alexseybaa92@yandex.ru (Y.V.B.); utvar@blagoslovenie.su (N.D.M.); noks59@yandex.ru (I.G.S.); bfvas@yandex.ru (B.F.V.); rea21@list.ru (E.A.R.); ovefr@yandex.ru (O.V.E.); 2Shemyakin-Ovchinnikov Institute of Bioorganic Chemistry of the Russian Academy of Sciences, 117997 Moscow, Russia; chai.mail0@gmail.com; 3Russian Entomological Society, 199034 Saint-Petersburg, Russia; lasius@yandex.ru

**Keywords:** ant nests, *Lasius niger*, microbial community, actinomycetes, bacilli, *Streptomyces antibioticus*-like, actinomycin D

## Abstract

In this study, the microbial communities of two nests of black garden ants *(Lasius niger*) in the hollows of stem branches of old apple trees were found to have similar species compositions: each community contained representatives of three species from the Bacillaceae family and one species of actinomycetes from the genus *Streptomyces*. In total, four types of bacilli and two actinomycetes were isolated. Actinomycetes were identified as *Streptomyces antibioticus*-like and *Streptomyces* sp. None of the bacilli had antibiotic activity, whereas both streptomycetes produced antibiotics that inhibited the growth of Gram-positive bacteria in vitro, including isolates from their community. Antibiotic compounds of *S. antibioticus*-like strain INA 01148 (Institute of New Antibiotics) were identified as actinomycin D and its closest homologue, actinomycin A. Actinomycins presumably change the microbial community of the ant nest substrate as they act against Gram-positive bacteria and against fungi and Gram-negative bacteria. The antibiotic activity of the isolated *Streptomyces* sp. INA 01156 is of interest, since the substances produced by this strain inhibit the growth of drug-resistant bacteria, including methicillin-resistant *Staphylococcus aureus* INA 00761 (MRSA) and vancomycin-resistant strain *Leuconostoc mesenteroides* VKPM B-4177 (VR) (VKPM–National Collection of Industrial Microorganisms (Russian acronym)).

## 1. Introduction

The study of natural microbial communities is of both general theoretical and descriptive interest and has significant application potential as microorganisms produce a large number of substances of value. In particular, most modern antibiotics are produced by microorganisms, but due to the increasing problem of resistance of pathogens to antibiotics, the search for new effective drugs must be continuous. To search for producers of new natural antibiotics, microorganisms can be isolated from insufficiently studied sources, for example, from bottom sediments of seas and freshwater ponds, from frozen soils, and from the symbionts of plants and animals—in this case, from ants and their habitats [1,2,3].

Ants are complexly organized social animals that are widespread across all continents (except Antarctica), living in many different conditions with different food resources. More than 14,000 species of ants have been described, and living species together account for as much as 10–25% of the biomass of all terrestrial animals [4]. However, only some species of ants have been studied for microorganisms on their integument, inside them, or in their nests. Most of the studies in this area have focused on leaf-cutting ants from the genera *Atta* and *Acromyrmex*, which live in the Western Hemisphere. These ants are also called fungus-growing ants, because they arrange fungi gardens for food in their nests, in which basidiomycetes from the family Agaricaceae are cultivated on shredded leaves. The productivity of such fungi decreases if the ascomycete fungi *Escovopsis* spp., which parasitize the fungi cultivated by the ants, settle in the nest. The success of these fungi is facilitated by the actinomycetes *Pseudonocardia* spp., which have been repeatedly found in ant nests. Representatives of the genus *Pseudonocardia* produce antibiotics that inhibit the growth of *Escovopsis* spp. [5,6,7,8,9,10].

The structure of one of the antibiotics produced by *Pseudonocardia* sp. associated with the fungus-growing ants *Apterostigma dentigerum* has been established. It is a dentigerumycin which is a cyclic depsipeptide with highly modified amino acids that selectively inhibits the associated parasitic fungus (*Escovopsis* spp.) [11,12]. The structures of the antibiotics gerumycins A–C produced by two different symbiotic *Pseudonocardia* spp. from ant nests are slightly smaller versions of dentigerumycin [13]. In addition to *Pseudonocardia* spp., symbionts of ants of other genera of actinomycetes have been isolated: *Amycolatopsis*, *Kitassatospora,*
*Kribbella*, *Propionicimonas, Streptomyces,* and *Tsukamurella* [7,14,15,16,17]. Known antibiotic antimycins A1–A4, valinomycins, and actinomycins, as well as macrolide antibiotic candicidins, have been identified from *Streptomyces* symbionts of leaf-cutting ants [18,19]. The authors of these publications suggested that actinomycetes impact the protection of ants or sanitation of the nest due to the formation of antibiotics.

Another series of publications deals with microorganisms associated with black garden ants (*Lasius niger*), which are widely distributed in Eurasia [20,21]. These ants feed on small invertebrates, as well as the sweet secretions of the aphids that they cultivate and protect [22]. In their anthills, representatives of the genus *Bacillus* dominate, and bacteria of other taxonomic groups, as well as actinomycetes of the genus *Streptomyces*, which, under laboratory conditions, exhibit antibiotic properties against bacteria from the same anthill, have also been found [23]. A comparative analysis of the abundance and biodiversity of actinomycete communities isolated from the living ants *L. niger* and *Formica cunicularia*, as well as their anthills, showed that the number of actinomycetes detected in *L. niger* ants is close to the number of actinomycetes in their anthills and the higher than that for *F. cunicularia*. The main actinomycetes were representatives of the genus *Streptomyces*, but *Micromonospora* and *Nocardia* were also found. The biodiversity of actinomycetes associated with living ants is considerably lower than in their nests and the surrounding soil [24,25].

The objectives of our study were to assess the diversity of microorganisms present in the nests of black garden ants (*L. niger*) and to explore their antibiotic properties.

## 2. Materials and Methods

### 2.1. Sampling of Substrates from the Ant Nests and Isolation of Microorganisms

The ant nests were located in the cavities of stem branches of two old apple trees in the Moscow Region, Russia. The trees were growing in a garden at a distance of 5 m from each other. The cavities were filled with a substrate resembling coarse-grained soil. Samples of the substrate were taken at the end of May 2016 in dry hot weather by hand using a sterile glove and placed in a sterile paper bag. Portions of 0.2 g of the substrates were added into 5 mL of sterile water, stirred in a shaker for 10 min, filtered through cotton, diluted, and plated on an agar medium within one week. After incubation at a temperature of 28 °C for 7 days, colonies of different types were seeded into tubes with a slanted agar medium. The reseeded clones were incubated for 1–4 days (bacteria) and 10–12 days (actinomycetes).

### 2.2. pH Determination of Substrates from the Ant Nests

To the weighed substrate, we added distilled water at a ratio of 1:5 before shaking for 1 h, filtered, and determined pH using a pH tester.

### 2.3. Media and Culture Conditions

For storage and maintenance of all microorganisms, modified agar Gause medium #2 was used (%): glucose 1, peptone 0.5, tryptone 0.3, NaCl 0.5, agar 2, and tap water (pH 7.2–7.4). The same medium without agar was used for the bacterial submerged cultivation and for the first stage of actinomycete cultivation. Liquid media for the second stage of actinomycete submerged cultivation were developed for strains producing antibiotics at the Gause Institute of New Antibiotics as described previously [26]. The composition of these media is indicated in %:

A (2663): glycerin 3, soy flour 1.5, NaCl 0.3, chalk 0.3, tap water (pH 7.0);

B (A4): glucose 1, soy flour 1, NaCl 0.5, chalk 0.25, tap water (pH 6.8);

C (Suc): sucrose 2, soy flour 1, NaCl 0.3, chalk 0.3, tap water (pH 6.8–7.0);

D (5339): glycerin 2, soy flour 0.5, (NH_4_)_2_SO_4_ 0.15, NaCl 0.3, chalk 0.3, tap water (pH 6.8);

E (6613): starch 2, corn extract 0.3, KNO_3_ 0.4, NaCl 0.5, chalk 0.5, tap water (pH 7.0–7.2);

F (330): sucrose 2.1, starch 0.85, pea flour 1.5, NaCl 0.5, NaNO_3_ 0.5, chalk 0.5, tap water (pH 7.0);

G (Am): sucrose 4, yeast extract 0.25, K_2_HPO_4_ 0.1, Na_2_SO_4_ 0.1, NaCl 0.1, (NH_4_)_2_SO_4_ 0.2, FeSO_4_∙7H_2_O 0.0001, MnCl_2_∙4H_2_O 0.0001, NaI 0.00005, chalk 0.2, distilled water (pH 6.5–6.7).

Submerged cultivation was carried out in 750 mL Erlenmeyer flasks with 150 mL of medium on a rotary shaker at 200 rpm at 28 °C. Inoculation with bacteria was conducted with a suspension of 10^7^ cells/mL of medium and antimicrobial activity in the culture liquid was determined after 1, 2, 4, and 7 days of growth. Actinomycetes in the first stage of cultivation were inoculated with a piece of agar medium into flasks with mycelium. After four days of growth, 7.5 mL of the culture grown in the first stage was transferred to flasks (1.5% v/v). The second stage of cultivation lasted 4–7 days. Bacterial test strains on the agar medium were grown at 37 °C with the exception of *Leuconostoc mesenteroides* VKPM B-4177 which was grown at 28 °C. Fungal test strains *Aspergillus niger* INA 00760 and *Saccharomyces cerevisiae* INA 01129 were grown at 28 °C for 48 and 24 h, respectively. 

### 2.4. Species Identification of Bacteria

#### 2.4.1. Morphological Characteristics

Bacteria were cultured on Gause medium #2. For the description of the actinomycetes’ cultural and morphological characteristics, Gause media #1, glycerin-nitrate agar and media ISP3, ISP4, and ISP5 were additionally used [27,28]. To describe the bacteria, the morphology of cells and spores was examined using a Micmed-6 light microscope (LOMO, Saint-Petersburg, Russia). For actinomycete species identification, characteristics such as the structure of the sporophores, the spore surface, the pigmentation of the air mycelium and substrate mycelium, and the pigment released into the medium were considered. Actinomycete spores were studied using a JEOL-6060A scanning electron microscope (JEOL, Tokyo, Japan) with a tungsten cathode. Before the study, platinum was sprayed onto the samples using a JFC-1600 apparatus (JEOL, Tokyo, Japan).

#### 2.4.2. Molecular Characteristics

The species of the isolated bacteria were determined via identification of the 16S rRNA gene sequences. Genomic DNA from the bacterial biomass was isolated using the PowerSoil DNA Kit (MO BIO, Carlsbad, CA, USA). PCR of the 16S rRNA gene was performed using a set of PCR Master Mix reagents (contains the Taq DNA polymerase; Thermo Scientific, Foster City, CA, USA) with universal bacterial primers: 27F (AGA GTT TGA TCC TGG CTCAG) and 1492R (TAC GGY TAC CTT GTT ACG ACT T) [29]. PCR was performed on a Thermal Cycler 2720 device (Applied Biosystems, Foster City, USA) according to the following program: (1) 94 °C for 5 min; (2) 30 cycles with temperature intervals of 94 °C for 1 min, 51 °C for 1 min, and 72 °C for 2 min; (3) 72 °C for 7 min. The nucleotide sequences were determined using the Sanger method on a Genetic Analyzer 3500 automatic sequencer (Applied Biosystems, Beverly, MA, USA) using universal bacterial primers: 27f, 341f (CCT ACG GGA GGC AGC AG), 1100r (GGG TTG CGC TCG TTG), and 1492r. The obtained sequences were aligned with the nucleotide sequences of the 16S rRNA gene of the bacterial type strains from the GenBank databases (blast.ncbi.nlm.nih.gov/Blast.cgi) and the Ribosomal Database Project (rdp.cme.msu.edu/) using the ClustalW program included in the MEGA7 program [30]. To construct phylogenetic trees, the neighbor-joining method was used, which is also included in the MEGA7 program. The statistical confidence of the branching order was established using bootstrap analysis of 10,000 alternative trees.

### 2.5. Determination of Antimicrobial Activity

To determine the antibiotic activity of microorganisms isolated from the ant nests, the following test strains were used: *Bacillus subtilis* ATCC 6633, *Bacillus mycoides* 537, *Bacillus pumilus* NCTC 8241, *Leuconostoc mesenteroides* VKPM B-4177 (a vancomycin-resistant strain, VR), *Micrococcus luteus* NCTC 8340, *Staphylococcus aureus* FDA 209P (methicillin-susceptible *Staphylococcus aureus*, MSSA), *St. aureus* INA 00761 (methicillin-resistant *Staphylococcus aureus*, MRSA), *Mycobacterium smegmatis* VKPM Ac 1339, *Mycobacterium smegmatis* mc^2^ 155, *Escherichia coli* ATCC 25922, *Comamonas terrigena* ATCC 8461, *Pseudomonas aeruginosa* ATCC 27853, *Aspergillus niger* INA 00760, and *Saccharomyces cerevisiae* INA 01129. Antibiotic activity was determined using the diffusion agar method. For this purpose, 100 μL aliquots of the tested solution were added to 9 mm diameter holes in the agar medium inoculated with the test strains. After incubation of 20–24 h, the diameters of the growth-inhibitory zones of the test strains were measured as an indicator of antibiotic action.

### 2.6. Isolation and Identification of Antibiotics

The supernatant of the *Streptomyces antibioticus*-like INA 01148 culture liquid was separated by centrifugation from the mycelium. Wet mycelia were extracted with ethanol at a neutral pH value. The supernatant was applied to Amberlyte XAD-2 (Serva Electrophoresis GMBH, Heidelberg, Germany) following elution by a mixture of n-butanol/acetone/water (1:1:1) at a neutral pH value. The eluate and the extract were dried at 37 °C and the residue was dissolved in methanol. The ratio of methanol concentrates to the volume of the original culture liquid was 1:100. The concentrates were analyzed by ultraviolet–visible (UV–Vis) spectrophotometry and thin-layer chromatography (TLC) on 10 × 10 cm^2^ DC-Alufolien Kieselgel 60 (Merck, Darmstadt, Germany) plates in a chloroform/methanol/water system (95:5:1). Antibiotic activity was determined by bioautography using test strain *St. aureus* INA 00761 (MRSA). The concentrates were initially separated by reversed-phase high-performance liquid chromatography (RP-HPLC) on an XBridge Ethylene Bridged Hybrid (BEH) column (C18 250 × 4.6 mm^2^, 5 μm, 130 Å) (Waters Corporation, Milford, MA, USA). Each sample (100 µL) was separated with a combined linear gradient of acetonitrile concentration from 16% to 95% solvent B (80% MeCN with the addition of 0.1% trifluoroacetic acid) relative to solvent A (0.1% trifluoroacetic acid) for 60 min. The flow rate was 950 µL/min and detection was monitored at 237 nm.

The samples were analyzed by ultra-performance liquid chromatography/mass spectrometry (UPLC-MS) using a Thermo Finnigan LCQ Deca XP Plus ion trap instrument with a Thermo Accela UPLC system (Thermo Fisher Scientific, Waltham, MA, USA) equipped with a YMC Triart microcolumn (C18 150 × 2 mm^2^, 1.9 μm) (YMC Co., Kyoto, Japan). Absorbance detection was monitored by an ultraviolet–visible diode array detector (UV-VIS DAD) (190–600 nm) and full scan mass spectrometry (MS) (electro spray ionization (ESI+), 150–2000 au). The samples were dissolved in a mixture of water/methanol/acetic acid (88:10:2) to a final concentration of 1 mg/mL, filtered through a 0.45 μm nylon filter, and injected into the liquid chromatography (LC) system.

## 3. Results

### 3.1. Identification of Ants and Microorganisms from the Substrate of the Ant Nests

Ant species were identified according to morphological characteristics (Figure 1) [20,21]. The black garden ant (*Lasius niger* (Linnaeus, 1758)) is the most common type of ant in Russia. Their colonies may be formed in small mounds, as well as under the bark and in the cavities of old trees, as in this case.

Gause agar medium #2, which is optimal for the growth of most cultivated bacteria and fungi according to our experience, was used to seed a suspension of substrates from the ant nests. The titers of colony-forming units (CFU) in the substrates of two ant nests are shown in Table 1.

Fungal colonies were completely absent in seeding. Four morphological types of bacterial colonies, including one type of actinomycete colonies, were found in each of the two substrates. About four to eight colonies of each type were removed into tubes with the agar medium. Three bacteria of each ant nest were Gram-positive, rod-shaped, and formed endospores, which indicated their belonging to the Bacillaceae family. Refinement of their taxonomic position and species identification were achieved using 16S rRNA gene sequence analysis (Table 2). The phylogenetic trees of six bacilli are shown in Figure 2.

Actinomycetes were identified by morphological characteristics and the 16S rRNA gene. Four actinomycete clones of nest 1 were slightly different from each other in terms of the color of the aerial mycelium (white, light gray, or gray), folding, and the diameter of the colonies. However, these differences were not fundamental, and these actinomycete isolates were characterized by common features such as short straight spore chains, smooth spore surface, gray aerial mycelium, substrate mycelium, and soluble pigment of yellow color when grown on Gause mineral agar medium #1 and of brown color on Gause agar medium #2. According to these characteristics, the species was identified as *Streptomyces antibioticus*-like. The sequences of the 16S rRNA gene of this actinomycete showed 100% similarity to the type strain *S. antibioticus* NRRL B-1701 (NRRL–Northen Regional Research Center) (Table 2, Figure 3). One of the actinomycete isolates was deposited in the culture collection of the Gause Institute of New Antibiotics (Russia) as *S. antibioticus*-like INA 01148.

Four actinomycete clones from nest 2 were characterized by a well-formed white aerial mycelium on Gause agar medium #2 or a white aerial mycelium with a light gray tint on Gause mineral agar medium #1, as well as colorless substrate mycelium; soluble pigment was not produced. This actinomycete had convoluted sporogenous hyphae and spores with characteristic spiky outgrowths (Figure 4). This description most closely matched the description of *Streptomyces cyanoalbus*, although the coincidence with the type strain of this species in the database was only 90.7% (Table 2, Figure 5) [31,32]. One actinomycete isolate was deposited in the collection as *Streptomyces* sp. INA 01156, and the study of its taxonomic affiliation will continue.

Thus, representatives of four species of the family Bacillaceae and two streptomycetes were isolated from the two nests of ants.

### 3.2. Antimicrobial Activity of the Bacterial Isolates

The antimicrobial properties of the six strains of four species of bacilli presented in Table 2 were determined. The activity of the whole bacterial broth culture was analyzed against 14 test strains of the microorganisms listed in Section 2. None of the isolated bacilli showed antibiotic activity against Gram-positive or Gram-negative bacteria or activity against fungi.

Four isolates of the *S. antibioticus*-like bacteria, including strain INA 01148, exhibited antibiotic activity against the Gram-positive bacterial test strains *St. aureus* FDA 209P, *St. aureus* INA 00761, *B. subtilis* ATCC 6633, and *M. luteus* NCTC 8340, as well as some antifungal activity in four media out of seven; activity against *Leuconostoc mesenteroides* VKPM B-4177 and *E. coli* ATCC 25922 was absent. There were no differences in the antimicrobial spectra of the four actinomycete isolates; the diameters of the zones of growth suppression differed insignificantly (Table 3). The effect of the broth of the isolates on each other was tested. *Bacillus* sp. INA 01161, *Bacillus muralis*-like INA 01162, and *Lysinibacillus pakistanensis*-like INA 01164 did not inhibit *S. antibioticus*-like INA 01148 growth, but *S. antibioticus*-like INA 01148 inhibited all three bacillary isolates (Table 3).

The antimicrobial activity of the broth of four *Streptomyces* spp. isolates from ant nest 2 was tested. No activity was detected in the two isolates. In another isolate (No. 4951), the activity of the broth was only found against *M. luteus* NCTC 8340 on two media out of seven (media A4 and 2663), and zones of growth inhibition were negligible (12–15 mm). Only one isolate was active in six out of seven culture media, *Streptomyces* sp. INA 01156 (Table 4). This isolate showed antimicrobial activity directed against Gram-positive bacteria only. Presumably, this strain can produce more than one antibiotic, because, for example, at approximately the same level of activity against *B. subtilis* ATCC 6633 and *M. luteus* NCTC 8340 (on media 6613 and 330), it also exhibited activity against *St. aureus* FDA 209P and *Leuc. mesenteroides* VKPM B-4177, but only on medium 330 (Table 4).

### 3.3. Isolation and Purification of Antimicrobial Substances from the Streptomyces antibioticus-like INA 01148 Broth

Since all four *S. antibioticus*-like isolates had the same antimicrobial spectrum of the broth during growth in different nutrient media and the concentrates had the same bright orange colorization, a single strain was taken for chemical analysis (*S. antibioticus*-like INA 01148). The antibiotic isolated had the following UV spectrum (MeOH): λ_max_ 217 nm (2.95 Abs), 237.2 nm (3.26 Abs), 426.6 nm (1.29 Abs), and 443 nm (1.30 Abs). According to TLC, the antibiotic had Rf = 0.7. Based on the UV spectrum and TLC data, the antibiotic matched the well-known antibiotic actinomycin D (dactinomycin). To confirm this conclusion, we undertook column calibration with the actinomycin D analytical standard (Serva, Catoosa, OK, USA). Three samples of the broth were analyzed by analytical RP-HPLC with absorbance detection at 237 nm: eluate supernatant concentrate (Figure 6A), mycelium extract concentrate (Figure 6C), combined eluate and supernatant concentrate after preparative TLC purification (Figure 6D), and actinomycin D standard (Figure 6B). As illustrated, all three *S. antibioticus*-like INA 01148 samples contained a major compound with a retention time of 42.6 min, which is quite close to the standard variant of actinomycin D.

For further identification, the combined eluate and supernatant concentrate after preparative TLC purification was analyzed by liquid chromatography–electrospray ionization–mass spectrometry (LC–ESI–MS), and an accuracy m/z value of the target compound was measured (Figure 7). As shown in Figure 7, the analyzed compound was represented by two approximately equal components with retention times (RT) of 7.34 and 7.54 min. The measured m/z value for the first peak (RT = 7.34 min) was 1255.6 Da, allowing identification of actinomycin D (calculated m/z value 1255.42 Da). The m/z value’s second peak (RT = 7.54 min) was 1269.47 Da (+ 14 Da), matching the peak of the highest homologue, actinomycin A1 [33].

### 3.4. Sensitivity of Bacterial Isolates to the Antibiotic Actinomycin D and the Interactions of These Isolates

After identifying actinomycin D as the antibiotic produced by the strain *S. antibioticus*-like INA 01148, we suggested that, in nature, this antibiotic can be produced and affect the growth of other microorganisms present in the substrate of ant nest 1 (Table 3). We found that all three bacilli were sensitive to small doses of actinomycin D (Table 5).

To simulate the effects of combined growth of bacteria in a substrate, we sowed four microbial isolates from ant nest 1 on the agar medium next to each other and incubated them at 28 °C. Six pairwise combinations of four different species isolates were examined (Table 2), but inhibition of the growth of any of the microorganisms due to the substances secreted into the agar medium by the neighboring strain was not observed. Since under submerged cultivation only *S. antibioticus*-like had antimicrobial properties, in the next step, we sowed *S. antibioticus*-like INA 01148 four days earlier than the three bacilli. With this seeding, the strain *S. antibioticus*-like INA 01148 inhibited the growth of three species of bacilli: *Lys. pakistanensis*-like INA 01164, *Bacillus* sp. INA 01161, and *B. muralis*-like INA 01162. The last two bacteria are shown in Figure 8.

## 4. Discussion

The production of antibiotics by microorganisms can be considered an evolutionarily developed defense mechanism in interspecific competition or a chemical weapon in the struggle for existence. A large number of different antibiotics produced by actinomycetes have been described, along with those produced by representatives of other groups of microorganisms such as bacilli and fungi.

Previous publications by various authors described the isolation of microorganisms from substrates of anthills. For example, the abovementioned fungus-growing ants and antibiotics produced by *Pseudonocardia* spp. These actinomycetes inhibit the growth of ant mycopathogens, such as *Escovopsis* spp. [18,19].

The symbiotic association between ants and antibiotic-producing microorganisms was previously reported by several different authors [5,6,7,8,9,10,12,13,14,15,19,23]. Various studies of microorganisms accompanying *L. niger* have been conducted, and fungi (from the genera *Penicillium, Acremonium,* and *Mucor*), bacteria of various taxonomic groups, including bacilli, and a large number of actinomycetes (from the genera *Streptomyces, Nocardia,* and *Micromonospora*) have been described [23]. The results obtained were unexpected for us, since fungi were completely absent in the cultures of substrates from the nests of ants we studied, and the species composition of the bacteria was poor: Gram-negative bacteria were absent and only four species of Gram-positive bacteria were found in the two nests. In both cases, they were representatives of two species of the genus *Bacillus* (*B. muralis*-like and *Bacillus* sp.), as well as *Lysinibacillus pakistanensis*-like (nest 1) and *B. aryabhattai*-like (nest 2); actinomycetes were represented by two species of the *Streptomyces* genus, one of which was identified as *S. antibioticus*-like (nest 1).

This small number of cultivated species cannot be explained by the selected agar medium for cultivation or by cultivation conditions. Traditionally, most species of bacilli, actinomycetes, and soil fungi isolated from the soils of the Moscow Region grow well on Gause agar medium #2. This medium is also suitable for the growth of microorganisms previously isolated from *L. niger* anthills [23,24,25]. Although the nests were located in the same garden in neighboring apple trees, the partial discrepancy in the species composition of bacteria can be explained by the difference in the properties of the substrates inside the nests (Table 1), which could be related to some differences in the trees and could affect the species composition of microbial communities. The concentrations of CFU in the substrates of both nests were approximately equal to each other and also fell into the range of concentrations of CFU found in previous publications on anthills of *L. niger* [23].

The results differ in terms of the number and diversity of microorganisms compared to those previously described for *L. niger*. This may be due to microbial communities in the anthills of *L. niger* located on the ground being more commonly studied. In this publication, we investigated the microbial communities localized in the cavities of old trees. The species composition of microorganisms from anthills varies depending on the time of year, being less diverse in spring than in autumn [23,34]. We took substrate samples from the ant nests in spring, which could have affected the species composition of the microbial community. Golichenkov et al. noted that seasonal changes in the bacterial complex in the domed part of the *Lasius niger* anthill may be associated with the antibiotic action of actinomycetes. This conclusion was drawn on the basis that, for example, absolute dominance of bacilli was observed in spring, when *Streptomyces albus* prevailed over actinomycetes, exhibiting a wide range of antibacterial activity against all bacteria isolated from the anthill (*Arthrobacter* spp., *Cellulomonas* spp., Flavobacterium-Cytophaga, and *Rhodococcus* spp.) with the exception of *Bacillus* spp. [23].

None of the four representatives of the Bacillaceae family showing antimicrobial activity was also unexpected, which, in our experience, is rare. On the contrary, both actinomycetes have antibiotic properties and, in addition to the test strains, also inhibit the growth of own bacteria (Table 3 and Table 4). Antibiotics produced by *S. antibioticus*-like INA 01148 were identified as actinomycin D and its analogue. In high concentrations, actinomycin D also inhibits the growth of fungi, which could explain the absence of fungi in the substrate of nest 1 [35,36]. Bacteria from nest 1 were sensitive to actinomycin D (Table 5), and this antibiotic probably inhibited bacterial growth when co-streaked on the agar medium, provided that the producer of actinomycin D was inoculated earlier and, probably, an antibiotic was already released into the medium (Figure 8). However, if the producer of actinomycin D and the bacilli are inoculated simultaneously, both strains of the bacilli grow and a part of the population forms spores. Under natural conditions in the microworld, bacilli likely also partially diverge with actinomycetes in time and space, which allows bacilli populations to survive. Actinomycin D producers among actinomycetes producing antibiotics are found in the nature at a high frequency—from 1:10 to 1:1000. In comparison, the frequency of occurrence in the nature of the producer of daptomycin is 1:1,000,000 [37]. This high frequency of occurrence in the nature is probably the reason that actinomycin D was one of the first discovered antibiotics [38]. Previously, actinomycins were isolated from substrates of nests, integuments, and feces of leaf-cutting ants of three species. Several antibiotics were discovered in that work, with the authors suggesting that the diversity of natural products plays a driving role in shaping the ecosystems of leaf-cutting ants [18].

*Streptomyces* sp. INA 01156 exhibited antibiotic properties against Gram-positive bacteria, including three species of bacilli, which were also isolated from ant nest 2 (Table 4). Further investigation of the structure of this antibiotic (or antibiotics) is planned.

The isolates of two populations of actinomycetes differed in the level of antibiotic activity (from zero to high). Such variability in natural actinomycete populations can have adaptive value when environmental conditions change if antibiotic formation is considered protection against competitive microflora.

## 5. Conclusions

The microbial communities of two nests of black garden ants *(Lasius niger*) in the hollows of stem branches of old apple trees showed similar species compositions: each community contained representatives of three species from the Bacillaceae family and one species of actinomycetes from the genus *Streptomyces*. Small differences in the species composition of microorganisms were apparently related to the difference in substrates in the nests, mainly in the pH value (7.81 and 7.22). This could affect the established species composition, although the trees were located at a distance of 5 m from each other.

None of the bacilli showed antibiotic activity, and both streptomycetes produced antibiotics that inhibited the growth of Gram-positive bacteria in vitro, including those from their community. We determined that one of the actinomycetes, *S. antibioticus*-like INA 01148, can produce actinomycins, due to the action of which Gram-negative bacterial and fungal microflora were suppressed. The second actinomycete, *Streptomyces* sp. INA 01156, can produce an antibiotic (or antibiotics) of an unknown nature that inhibits the growth of drug-resistant bacteria, which may be of practical interest.

The difference between these microbial communities and those previously described in *L. niger*, and represented by a large number of species of actinomycetes, fungi, and bacteria of different taxonomic groups, is probably due to anthills located on the ground and not ant nests being studied.

## Figures and Tables

**Figure 1 life-10-00091-f001:**
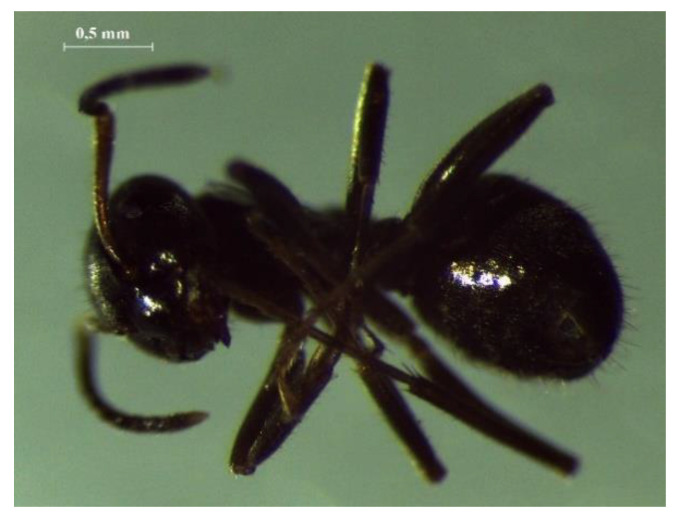
Black garden ant *Lasius niger*.

**Figure 2 life-10-00091-f002:**
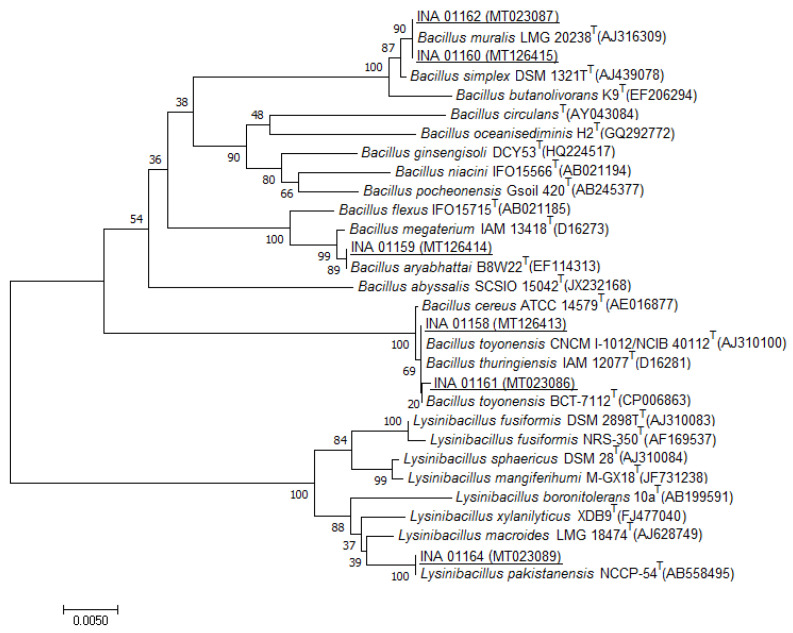
Phylogenetic position of the strains INA 01158–INA 01162 and INA 01164 based on the results of comparative analysis of the 16S rRNA gene nucleotide sequences. The scale corresponds to five base substitutions for every 1000 nucleotides (evolutionary distance). INA–Institute of New Antibiotics, LMG–Collection of the Laboratorium voor Microbiologie en Microbiele Genetica, DSM–Deutsche Sammlung von Mikroorganismen und Zellkulturen, IAM–Institute of Applied Microbiology, SCSIO–South China Sea Institute of Oceanology, ATCC–American Type Culture Collection, and CNCM–Collection Nationale de Cultures de Microorganismes.

**Figure 3 life-10-00091-f003:**
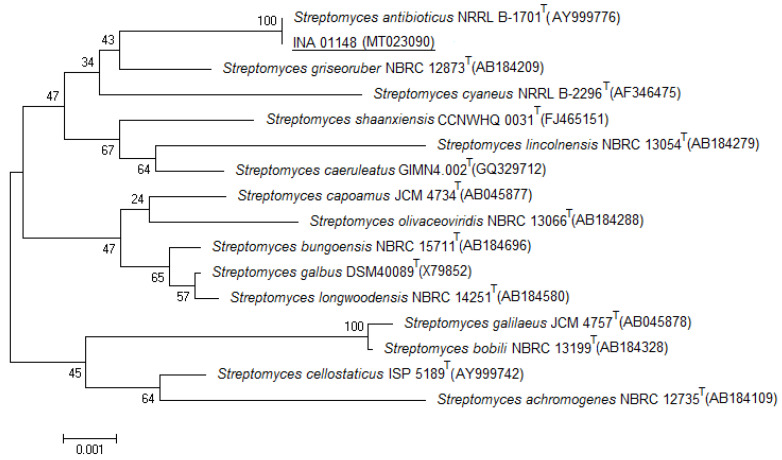
Phylogenetic position of the strain *Streptomyces antibioticus*-like INA 01148 based on the results of a comparative analysis of the 16S rRNA gene nucleotide sequences. The scale corresponds to one base substitution for every 1000 nucleotides (evolutionary distance). NRRL–Northen Regional Research Center, NBRC–Biological Resource Center, JCM–Japan Collection of Microorganisms, and ISP–International *Streptomyces* Project.

**Figure 4 life-10-00091-f004:**
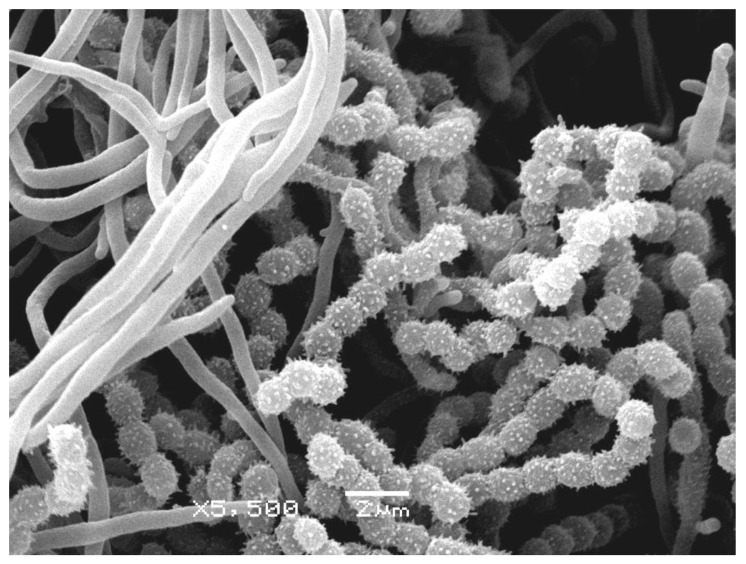
Vegetative mycelium, sporogenous hyphae, and spores of *Streptomyces* sp. INA 01156 (scanning electron microscopy).

**Figure 5 life-10-00091-f005:**
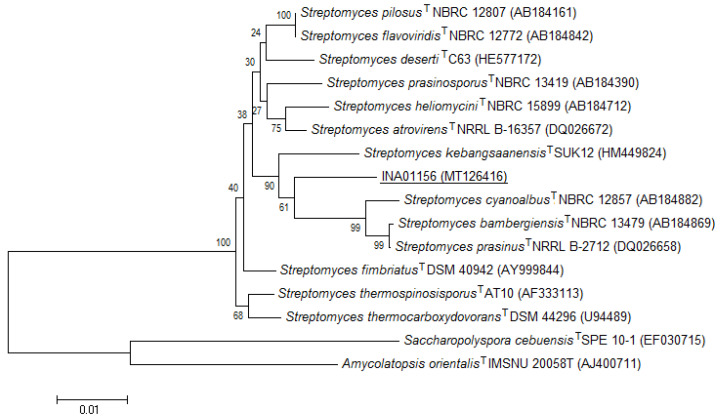
Phylogenetic position of the strain *Streptomyces* sp. INA 01156 based on the results of comparative analysis of the 16S rRNA gene nucleotide sequences. The scale corresponds to two base substitutions for every 1000 nucleotides (evolutionary distance). IMSNU–Institute of Microbiology, Seoul National University.

**Figure 6 life-10-00091-f006:**
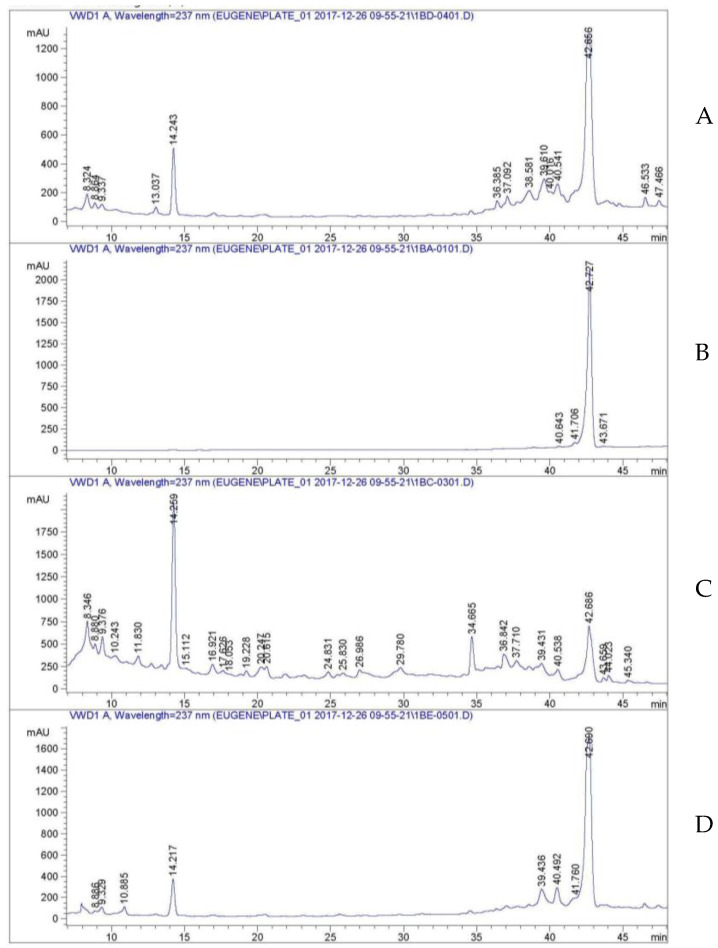
Comparative analytical reversed-phase high performance liquid chromatography (RP-HPLC) analysis of the eluate supernatant concentrate (**A**), the mycelium extract concentrate (**C**), and the combined eluate and supernatant concentrate after preparative thin-layer chromatography (TLC) purification (**D**) against the actinomycin D analytical standard (**B**).

**Figure 7 life-10-00091-f007:**
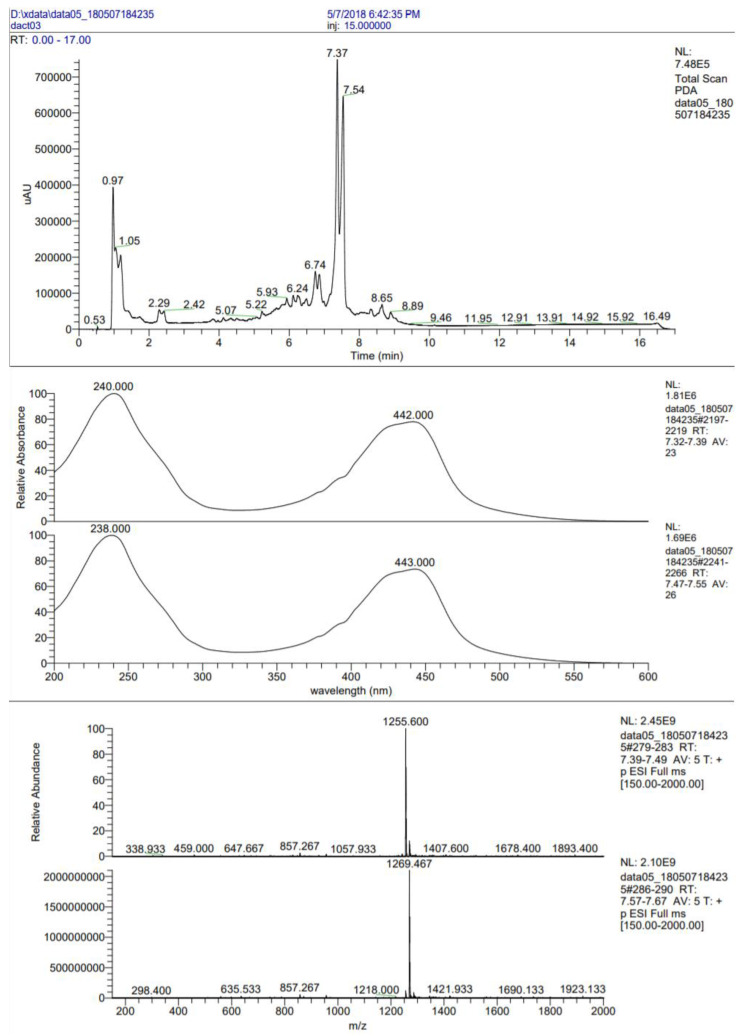
Liquid chromatography–electrospray ionization–mass spectrometry (LC–ESI/MS) analysis of the sample with two actinomycin components after previous preparative thin-layer chromatography (TLC) purification: the top panel shows the profile of the analytical reversed-phase HPLC of the sample (actinomycin D is eluted at 7.37, actinomycin A at 7.54 min, respectively); the middle panel shows the UV spectra of actinomycin D and actinomycin A (top–down); the lower panel shows the ESI/MS spectra of actinomycin D and actinomycin A (top–down). RT–retention time; NL–intensity of the base peak; PDA–type of measurement; AV–mode of measurement (average).

**Figure 8 life-10-00091-f008:**
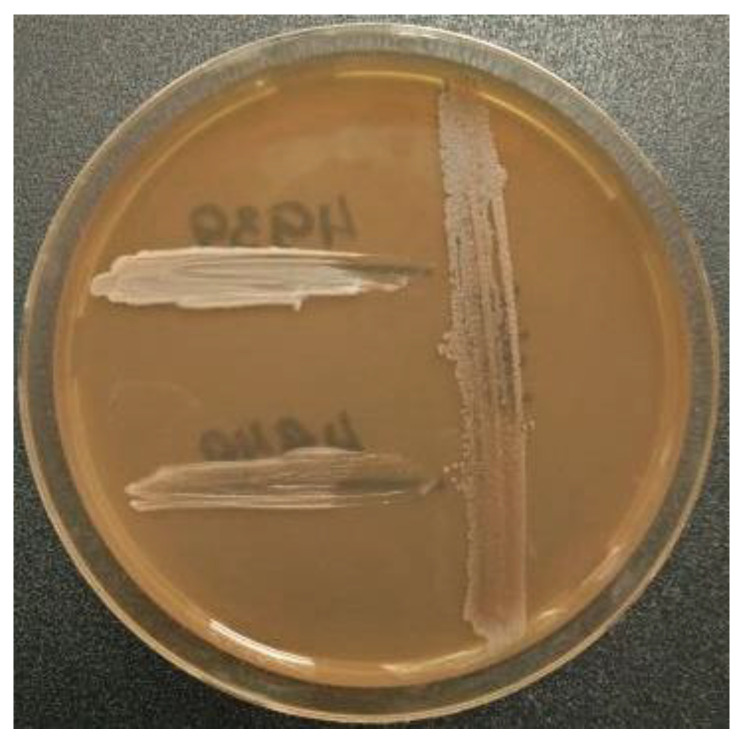
*S. antibioticus*-like INA 01148 (vertical bar) inhibits the growth of the bacteria *Bacillus* sp. INA 01161 (top bar) and *B. muralis*-like INA 01162.

**Table 1 life-10-00091-t001:** General characteristics of two ant nests.

Ant Nests	Apple Variety	Substrate in Ant Nests		
		Color	pH	CFU/g *
1	Antonovka	Brown	7.81 ± 0.08	1.3 × 10^5^
2	Korichnoe	Black	7.22 ± 0.13	8.2 × 10^4^

* colony forming units/g (CFU).

**Table 2 life-10-00091-t002:** Species identification of microorganisms from ant nests of *Lasius niger* by 16S rRNA gene analysis.

Ant Nests	Species, Strains	DNA (bp)	Percent Identity	Accession Numbers of the Deposited Sequences (GenBank)
1	*Bacillus muralis*-like INA 01162	1393	98	MT023087
	*Bacillus* sp. INA 01161	1437	99.5	MT023086
	*Lysinibacillus pakistanensis*-like INA 01164	1403	100	MT023089
	*Streptomyces antibioticus*-like INA 01148	1299	100	MT023090
2	*Bacillus aryabhattai*-like INA 01159	1405	99.5	MT126414
	*Bacillus muralis*-like INA 01160	1396	98.7	MT126415
	*Bacillus* sp. INA 01158	1431	100	MT126413
	*Streptomyces* sp. INA 01156	1373	90.7	MT126416

**Table 3 life-10-00091-t003:** Antimicrobial activity of the *Streptomyces antibioticus*-like INA 01148 broth against test strains and bacterial isolates from ant nest 1 (zones of growth inhibition in mm).

Culture Media [26]	Collection Test Strains								Bacillary Isolates from Ant Nest 1		
	*Staphilococcus aureus* INA 00761	*St. aureus* FDA 209P	*Bacillus subtilis* ATCC 6633	*Micrococcus luteus* NCTC 8340	*Leuconostoc mesenteroides* VKPM B-4177	*Escherichia coli* ATCC 25922	*Aspergillus niger* INA 00760	*Saccharomyces cerevisiae* RIA 259	*Bacillus* sp. INA 01161	*B. muralis*-like INA 01162	*Lys. pakistanensis*-like INA 01164
2263	24.3 ± 0.6	23.7 ± 1.5	23.3 ± 2.3	22.3 ± 1.5	0	0	0	0	25.3 ± 2.5	30.7 ± 3.1	22.3 ± 3.8
A4	23.3 ± 5.9	20.7 ± 4.0	24.0 ± 6.9	23.0 ± 6.1	0	0	14.3 ± 0.6	19.7 ± 1.5	26.7 ± 5.1	33.3 ± 3.1	29.7 ± 4.9
Suc	27.3 ± 3.1	25.3 ± 1.5	27.3 ± 2.5	24.7 ± 3.1	0	0	0	0	29.3 ± 3.2	31.7 ± 1.2	29.7 ± 0.6
5339	23.3 ± 4.7	21.0 ± 5.3	23.7 ± 4.2	22.0 ± 4.0	0	0	0	17.7 ± 3.1	24.7 ± 4.5	30.7 ± 3.5	20.3 ± 2.5
6613	26.0 ± 1.0	24.0 ± 3.6	26.3 ± 2.1	23.3 ± 4.7	0	0	0	13.7 ± 1.5	30.0 ± 2.0	30.0 ± 2.6	32.3 ± 3.1
330	27.3 ± 1.2	24.0 ± 3.5	26.0 ± 2.6	25.3 ± 1.5	0	0	0	13.7 ± 1.5	24.7 ± 2.1	32.7 ± 2.5	32.0 ± 3.6
Am	25.7 ± 2.5	22.7 ± 4.2	24.7 ± 4.0	24.7 ± 3.1	0	0	0	0	24.7 ± 2.1	31.3 ± 3.1	30.3 ± 1.5

**Table 4 life-10-00091-t004:** Antimicrobial activity of the *Streptomyces* sp. INA 01156 broth against test strains and bacterial isolates from ant nest 2 (zones of growth inhibition in mm).

Culture Media [26]	Collection Test Strains								Bacillary Isolates from Ant Nest 2		
	*St. aureus* INA 00761	*St. aureus* FDA 209P	*B. subtilis* ATCC 6633	*M. luteus* NCTC 8340	*Leuc. Mesenteroides* VKPM B-4177	*E. coli* ATCC 25922	*A. niger* INA 00760	*Sac. cerevisiae* RIA 259	*Bacillus* sp. INA 01158	*B. muralism*-like INA 01160	*B. aryabhattai*-like INA 01159
2263	0	0	0	0	0	0	0	0	0	0	0
A4	0	0	0	17.3 ± 2.5	0	0	0	0	18.7 ± 3.1	15.3 ± 3.2	19.7 ± 2.9
Suc	0	0	0	15.7 ± 3.6	0	0	0	0	17.3 ± 3.0	18.7 ± 4.2	19.7 ± 1.1
5339	0	0	17.3 ± 4.0	19.3 ± 3.1	0	0	0	0	20.3 ± 1.5	20.7 ± 3.1	21.3 ± 1.5
6613	0	0	21.7 ± 2.1	21.7 ± 2.5	0	0	0	0	21.0 ± 2.3	20.0 ± 2.7	20.3 ± 3.4
330	0	15.7 ± 2.5	20.7 ± 3.5	19.7 ± 2.1	15.3 ± 0.6	0	0	0	19.7 ± 3.1	20.7 ± 2.6	22.0 ± 0.6
Am	15.3 ± 1.5	15.3 ± 0.6	12.3 ± 0.6	18.7 ± 3.5	0	0	0	0	16.7 ± 2.7	17.3 ± 3.1	20.1 ± 1.2

**Table 5 life-10-00091-t005:** Minimum inhibitory concentration (MIC) of actinomycin D for inhibiting the growth of isolates from the substrate of the ant nest 1 strains.

Microbial Isolates from Ant Nest	MIC of Actinomycin D (µg/disc)
*Bacillus muralis*-like INA 01162	0.125
*Bacillus* sp. INA 01161	0.06
*Lysinibacillus pakistanensis*-like INA 01164	0.125
*Streptomyces antibioticus*-like INA 01148	>64 *

* The maximum concentration in the experiment.

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
