# Peer review of "Antimicrobial Activity of Microorganisms Isolated from Ant Nests of *Lasius niger"

_life, 2020, doi:10.3390/life10060091_

Round 1
Reviewer 1 Report
The manuscript by Efimenko et al describes the analysis of the microbiota of two nests of black garden. The isolates got from the nests are characterized and identified by molecular methods. Besides, the antimicrobial properties of the isolated strains are studied. One of the actinomycete strains produced a known antibiotic, actinomycin D, and the other strain produces an unknown molecule yet.
The experiments are well conducted, and the methodology used is adequate and well described. The bibliography is correct, and a good discussion of the results is made.
The paper is of interest from the point of view of the discovery of new antimicrobial molecules and also, from the ecologically point of view. The strategies adopted by ants to avoid infection and to compete for the nutrients are astonishing.
Therefore, I recommend it for publication
Author Response
Dear reviewer,
Thanks for the detailed analysis of the manuscript.
Yours faithfully,
Dr. T.A. Efimenko
Dr. O.V. Efremenkova
Reviewer 2 Report
The revised version of the work presented by Efimenko et al. has been extensively improved according to several previous comments, so I recommend its publication in the present form.
Line 143: please specify that Mega7 is a program: "MEGA7 program [30]"
Author Response
Dear reviewer,
Thanks for the detailed analysis of the manuscript. We have made changes to the text.
Yours faithfully,
Dr. T.A. Efimenko
Dr. O.V. Efremenkova
Reviewer 3 Report
I have only a few minor comments.
Line 33-35 – The usage of ‘continual’ is not suitable here which means ‘a process over the duration of time with interruptions’. I suppose the authors want to use ‘continuous’. Please rectify it.
Line 49 – Remove ‘agriculture’.
Line 216 – Change to ‘One of the actinomycete isolates was…’
Line 246 – Is the activity tested using filtered broth or the whole bacterial-broth culture?
Line 263-264 – Change to “Only one isolate was active in six out of seven culture media, Streptomyces sp. INA 01156 (Table 4).
Line 274, 276, and 285– Please italicize the species.
Line 316 – What is ‘joint growth’? Do you mean ‘combined growth’?
Line 321 – Remove ‘out’ after ‘sowed out’
Line 329 – Add ‘as’ after ‘considered’
Line 338-339 – Change to “The symbiotic association between ants and antibiotic-producing microorganisms was previously reported by several different authors”. It requires citations also.
Author Response
Dear reviewer,
Thank you for your detailed analysis of the manuscript and your valuable comments. We have taken into account all the comments made.
Yours faithfully,
Dr. T.A. Efimenko
Dr. O.V. Efremenkova
This manuscript is a resubmission of an earlier submission. The following is a list of the peer review reports and author responses from that submission.
Round 1
Reviewer 1 Report
The manuscript Efimenko et al reports the bacterial community present in nests of Lasius niger, in order to find new compounds useful for biomedical purposes.
The work is well designed and perfomed although the results are not as sound as would be desired. The antibiotics produced by S. antibioticus are already known (actinomycin A and D) and the compound produced by S. cyanoalbus has not been determined.
Nevertheless, the wok is interesting for the research community in order to understand the ecology of the Lasius niger nests and the way they protect their colonies against enemmies. The discussion is wide and well structured.
Some minor comments:
Line 44: Introduce "are produced by": ... most modern antibiotics are produced by...
In results, past tense should be used in general.
For example in line 172: !....nests did not differ ..."; line 178: ...The formed...".......line 209...
The first letter of Gram should be in capital letter . line 177, line 210, 212...
In line 227 a reference to Table 4 should be added
In line 292 should be added an epigraph of "Discussion".
In the references the microorganisms should be in italics.
I consider the paper suitable for publication
Yours sincerely
Author Response
Dear Reviewer,
Thank you very much for your time and help in preparing the article. We took all the comments into account. Regarding the remark, I would like to clarify: “In line 227 a reference to Table 4 should be added.” On the recommendation of another reviewer, we included in the section Methods compositions of media, there is also a reference.
Yours sincerely,
Dr. Tatiana A. Efimenko
Dr. Olga V. Efremenkova
Reviewer 2 Report
The work presented by Efimenko et al. describes the microbial diversity and antibiotic production in two ant nests of Lasius niger. To achieve this goal, appropriate experimental procedures and analyses have been employed. Thus, after cultivation procedures, different bacteria were recovered and the production of antibiotics was observed. The study overall is very interesting, but I cannot recommend accepting this manuscript in the present form.
Broad comments
The text in general needs further English proofreading because it’s difficult to follow.
The molecular identification of bacteria by using the 16S gene needs further analyses. A phylogenetic tree may be included to better assess the identification of the strains recovered.
Specific comments
Title: The current title does not mention the work carried out on antibiotic production. I suggest the following title: “Microbial diversity and antibiotic production in ant nests of Lasius niger”.
Abstract: It’s too long and needs to be reduced.
Abstract
Line 14: remove “By”
Line 16: “On the basis of”
Line 17: All four types of what? Please check this sentence.
Lines 21 and 22: Please replace “have only isolates of two actinomycetes species.” By “was only observed in the two actinomycetes species”
Line 23: Please, remove “were”
Lines 25-26: This sentence is difficult to understand. Please, check it.
Line 32: Please replace “since produced substances inhibit” by “since it inhibits”.
Lines 34-37: This sentence is too long. Please, split it up into two sentences.
Introduction
Line 44: Replace “practically significant for people” by “of interest”.
Line 44: “most modern antibiotics are produced by microorganisms”.
Line 65: “inhibits the growth of”.
Lines 79-83: This sentence is too long. Please, split it up into two sentences.
Line 84: “were also found”.
Line 87: “were the diversity of microorganisms”.
Materials and Methods
Line 94: Please replace “with a hand in a sterile glove” by “by hand using a sterile glove”.
Line 103: “Media and culture conditions”.
Line 118 “,respectively”.
Line 119: “Molecular identification of bacteria”.
Line 120: Please describe the “morphological characters” employed for bacterial identification in a different section and place before “molecular identification”.
Line 125: Please, provide the references for these two primers.
Line 126: “according to the following program”.
Lines 126-127: Please, replace the hyphen by "for".
Line 128: “on an automatic”.
Line 130: “and 1492”.
Line 130: Were sequenced on both strands? Please, clarify.
Lines 131-133: Please, provide more details on how the analysis was performed and the parameters used (i.e. e-value, etc).
Line 139: “the following test strains”
Line 146: Please, provide more details on this test as this paragraph is merely a list of microorganisms.
Line 148: “centrifugation from the mycelium”
Lines 153-160: Please provide the names for the following acronyms: UV-VLS, TLC, YMC, UV-VIS DAD, MS and LC.
Results
Line 172: If a statistical test was not performed, please remove the word "significantly"
Line 177: ”eubacteria were”.
Line 178: This is not correct as there are other bacteria capable of producing endospores i.e members of the family Clostridiaceae. Please, clarify it.
Line 183: “isolates were”.
Line 188: Which collection?
Line 190: “were characterized”.
Line 200: Table 2 lacks relevant information. Please provide further details: i.e. e-value. Also, scientific names and the GenBank accession numbers of their closest relatives.
Lines 209-210: I do not understand this sentence. Do you mean that there is no antibiotic activity at all?
Line 213: Please, list these media in the M and M section.
Line 216: By productivity, do you mean antibiotic production? Please clarify.
Lines 217-219: “which suggests that on different media one antibiotic or one complex of antibiotics that are similar in chemical structure is formed by these strains”: This should better go to discussion, but I don't see what led you to that conclusion.
Lines: 225-225: In Table 3:Where is Streptomyces antibioticus INA 01148 among the isolates tested from the ant nest? Please clarify this point.
Line 228: where is this isolate listed? Please, clarify this.
Line 242-243: In Table 4: Where is Streptomyces cyanoalbus INA 01156 among the isolates tested from the ant nest?. Please clarify this point.
Line 253: “Three samples”.
Line 266: I think here you mean “respectively” instead of “relatively”. Please clarify.
Lines 278-279: In Table 5: What's mcg? Please, clarify it.
Line 288: Figure 5. What about Lysinibacillus pakistanensis in this experiment?
Discussion
Line 292: From here (until Conclusion) these paragraphs should be included in a discussion section. Therefore, please, head it as Discussion.
Lines 313-320: The presence of more bacteria and fungi may be revealed by culture-independent approaches. The lack of this sort of procedures may be a limitation of the present study. Consequently, more discussion on this aspect is needed here.
Lines 324-327: This sentence is too long. Please, split it up into two different sentences.
Line 337: This information is not present in those tables. Please, clarify it.
Lines 338-339: Reference needed.
Line 350: Please, format this citation properly.
Line 355: Please, replace “Clarification” by “Further investigation”.
Conclusion
Line 370: This result is not shown.
Line 375-378: This sentence is too long; please, try to split it up into two sentences.
References
Please, check the scientific names as some them should be written in italics
Author Response
Dear Reviewer,
Thank you for carefully reading the manuscript and the comments made. They were very useful for improving the manuscript. English is not our native language, but we did not suspect that the text was so bad.
We added phylogenetic trees.
The title of the article was changed, although we wanted to emphasize the interactions between bacteria in the community. Apparently, this did not work, and the main focus was on antibiotics. Another reviewer made the same comment.
The diffusion method in agar was inserted into the Methods section.
Abbreviations UV-VLS, TLC, YMC, UV-VIS DAD, MS and LC are decoded.
Thanks for the comment on endospores.
Since we introduced phylogenetic trees, we did not start adding related strains from the database to table 2.
«Lines 217-219: “which suggests that on different media one antibiotic or one complex of antibiotics that are similar in chemical structure is formed by these strains”: This should better go to discussion, but I don't see what led you to that conclusion» We have removed this paragraph, it needs to be explained in great detail.
«Lines: 225-225: In Table 3: Where is Streptomyces antibioticus INA 01148 among the isolates tested from the ant nest? Please clarify this point»: Antibiotic activity was determined in four isolates. It was approximately the same in all four isolates. The table shows the antimicrobial spectrum of one of these isolates that was deposited in the INA collection. The same table in the three extreme right columns shows the activity of this actinomycete against eubacteria isolated from the same microbial community.
«Line 242-243: In Table 4: Where is Streptomyces cyanoalbus INA 01156 among the isolates tested from the ant nest?. Please clarify this point»: Same as with Streptomyces antibioticus. In the text (now these lines are 278-292), three isolates are indicated, the fourth is presented in the table. The same table in the three extreme right columns shows the activity of this actinomycete against eubacteria isolated from the same microbial community.
«Line 288: Figure 5. What about Lysinibacillus pakistanensis in this experiment?»: In all three bacteria, when seeding after 4 days on agar medium near S. antibioticus INA 01148, growth was suppressed. Only two strains were shown in the photo: B. toyonensis INA 01161 and B. muralis INA 01162. Therefore, there is an error in the text. Tables 3 and 5 show that all three bacteria (including Lysinibacillus pakistanensis) are sensitive to actinomycin and to S. antibioticus culture liquid INA 01148.
«Lines 313-320: The presence of more bacteria and fungi may be revealed by culture-independent approaches. The lack of this sort of procedures may be a limitation of the present study. Consequently, more discussion on this aspect is needed here»: Yes, it is possible to evaluate unculturable microorganisms in a sample. We did not set such a task, we compared the results with previous publications, which also described the microorganisms that were seeded on agar medium.
“Line 337: This information is not present in those tables. Please, clarify it.»:
This information is provided in the text:
"3.2. Antimicrobial activity of microorganism isolates in submerged cultivation
The antimicrobial properties of the six strains of four species bacilli presented in Table 2 were determined. The activity of the cultural liquid was analyzed against 14 test strains of the microorganisms listed in the Methods section. None of the isolated bacilli showed antibiotic activity against Gram-positive and Gram-negative bacteria, as well as fungi"
“Line 370: This result is not shown.” : This could affect the established species composition, although the trees are located at a distance of 5 m from each other.
This information is provided in the Methods section (line 82).
Thank you very much for your help.
Yours faithfully,
Dr. Tatiana A. Efimenko
Dr. Olga V. Efremenkova
Reviewer 3 Report
Brief summary
In this study, the authors isolated bacteria belonging to the family Bacillaceae and Streptomycetes from two nests of ants-Lasius niger using the traditional culture-dependent methodology. These species were tested for their ability to produce antibiotics
Broad comments
Strength: The main strength of this article is the authors’ attempt to find new antibiotics from the ant-associated microbes.
Weakness: The manuscript has severe flaws. Firstly, it is poorly structured and written. It is very difficult to understand the authors’ interpretations and discussion, mostly due to poor sentence construction. Please see my following comments for more details (Note that I haven’t mentioned all of them since this manuscript requires rigorous reformatting). I only included my comments until some sections since it has to be extensively rewritten.
Specific Comments
Line 2- The title of the article does not correspond to the research performed. A valid and more sensible title is needed.
Lines 11-37 – Please restructure the abstract. Many parts of the abstract are not clear. I recommend the authors to shorten this section for better readability.
Lines 14-16 – “By three morphological types of Gram-positive bacterial colonies and one type of actinomycetes colonies were observed in each of two nests.” It is not clear what authors want to say here. Revise it.
Lines 16- 18- Rephrase the sentence to “Based on morphology and 16S rRNA gene sequences 16 bacterial types belonging to the family Bacillaceae: Bacillus aryabhattai, B. 17 muralis, B. toyonensis and Lysinibacillus pakistanensis were identified.
Line 44 - Change to ‘In particular, most modern antibiotics are produced by microorganisms,…’
Lines 76-79 – Please restructure the sentence for clarity.
Lines 87-88 – Restructure the sentence to “The objectives of our study were to assess the diversity of microorganisms present in the nests of the black garden ant (L. niger), and their antibiotic properties.
Line 93 – 94 – Change to “Samples of the substrate were taken at the end of May 2016 in dry, hot weather wearing a sterile glove and placed in a sterile paper bag”
Line 95- Change to “Portions of 0.2 g of the substrates were added into 5 ml of sterile water”….
Lines 97 – Insert a comma after “7 days..”
Line 104 – Insert a comma after “microorganisms…”
Line 110 – Remove “a temperature of”
Lines 120-121 – Change to “The bacterial isolates were determined by morphological characters, as well as using 16S rRNA gene sequencing”
Lines 120 -133 – Add citation for 27F/1492 R and 341F/1100r primer pairs. What was the advantage of using two different sequencing primers? What was the total length obtained after sequencing the 16S rRNA gene?
Where were new sequences deposited?
Line 151 – Please clarify “The eluate and extract were evaporated to dryness”. Does it mean “to dry”?
Line 172 – Haven’t done any statistical tests to prove the significant differences. So please remove the word “significant” from the sentence.
Lines 178-179 – Change to “Species were determined by using 16S rRNA gene sequence analysis (Table 2).”
Lines 187-188 – Change to “The sequences of the 16S rRNA gene of this actinomycete showed 100% similarity to the type strain S. antibioticus (Table 2).
Author Response
Dear Reviewer,
Thank you very much for the detailed analysis of the manuscript. We have taken into account your comments.
We have reduced the abstract.
Name changed.
Thank you for taking the time to help us improve the text.
We present the new version at your discretion.
Yours sincerely,
Dr. Tatiana A. Efimenko
Dr. Olga V. Efremenkova
Round 2
Reviewer 2 Report
The revised version of the work presented by Efimenko et al. has been improved according to several previous comments. I recommend accepting the manuscript after the following revisions.
- Please remove the comma from the title.
- In the materials and methods section please include the program used for phylogenetic analyses and some details on how these analyses were performed.
- Table 2. This table still lacks information about 16S-rRNA sequences as percent identity is not enough. This table should include the scientific names, E-values and the GenBank accession numbers of their closest relatives.
Author Response
Dear Reviewer,
Thank you again for your attention to our article. The title was corrected and information about tree construction was added.
About entering information in table 2: E-values for all isolates – 0. In addition, we have included trees in the article, which shows the relationship with the type strains. We think that additional information is not necessary, because it will make the table heavier.
Thank you very much for your help.
Yours faithfully,
Dr. Tatiana A. Efimenko,
Dr. Olga V. Efremenkova
Reviewer 3 Report
I completely understand that English is not the native language of the authors and appreciate the authors’ effort to improve it throughout the manuscript.
Lines 87-88- Haven’t followed my previous suggestions.
In the previous review report, I suggested “Restructure the sentence to “The objectives of our study were to assess the diversity of microorganisms present in the nests of the black garden ant (L. niger), and their antibiotic properties.” If authors are reluctant to follow my previous comment, consider changing to “The objects of our study were to determine the microorganisms present………”
Lines 127 – change to ‘characters such as ……..’
Figure 2- Please follow the taxonomic nomenclature rules for the species described in the phylogenetic tree. Several tree editing software such as figtree (http://tree.bio.ed.ac.uk/software/figtree/), treeview are available for this task.
Figure 3- Same comment as above.
Figure 5- Same comment as above
Line 253 – Capitalize the first letter of “streptomycetes”
Author Response
Dear Reviewer,
Thank you for your careful reading of our manuscript and your comments.
For lines 87-88, 127: we have made changes, thank you for your comment.
For figures 2, 3, 5: trees were corrected, thank you for the link to the program.
For line 253: There is no name for the genus Streptomyces, but only the concept of "streptomycetes", as well as "bacteria" or " actinobacteria".
Thank you very much for your help and attention to our article.
Yours sincerely,
Dr. Tatiana A. Efimenko
Dr. Olga V. Efremenkova